# Characterizing the Effect of Noise in Language Generation in the Limit

**Aaron Li** [* 1]   **Ian Zhang** [* 2]

## Abstract

Kleinberg and Mullainathan recently proposed a formal framework for studying the phenomenon of language generation, called *language generation in the limit*. In this model, an adversary gives an enumeration of example strings from an unknown target language, and the algorithm is tasked with correctly generating unseen strings from the target language within finite time. Refined notions of non-uniform and uniform generation were later introduced by Li, Raman, and Tewari (2025), and a noisy model was introduced by Raman and Raman (2025), which allows the adversary to insert extraneous strings. A natural question in the noisy model is to quantify the effect of noise, by studying the impact of each additional extraneous string. We show two complementary results in this setting. We first show that for both uniform and non-uniform generation, a single noisy string strictly reduces the set of collections that can be generated, thus answering an open question in Raman and Raman (2025). Then, we show for both uniform and non-uniform generation that generation with a single noisy string is equivalent to generation with any finite amount of noise, sharply contrasting with the strict hierarchy for noisy generation in the limit shown by Bai, Panigrahi, and Zhang (2026). Finally, we leverage our previous results to provide the first known characterization for non-uniform noise-dependent generatability.

## 1. Introduction

Understanding the powers and limitations of Large Language Models (LLMs) is a fundamental problem in machine learning. A wide range of recent work has explored various aspects of this technology, including mitigating

---
[*]Equal contribution   [1]Harvard University [2]Duke University. Correspondence to: Aaron Li <aaronli@college.harvard.edu>, Ian Zhang <ian.zhang@duke.edu>.

*Proceedings of the 43$^{rd}$ International Conference on Machine Learning*, Seoul, South Korea. PMLR 306, 2026. Copyright 2026 by the author(s).

hallucinations (Kalai & Vempala, 2024; Kalai et al., 2025), and uncovering internal world-models (Vafa et al., 2024; Li et al., 2023). Another important line of research has focused on understanding the impact of noisy training data on the quality of LLM outputs. Many empirical studies have shown the degradation of model performance under noisy labels, and have proposed various strategies to increase robustness to noise (Han et al., 2018; Ren et al., 2018; Li et al., 2020). Within statistical learning theory, there has also been great interest in understanding the impact of noise on learning (Angluin & Laird, 1988; Kearns & Li, 1993; Natarajan et al., 2013).

In this paper, we study the problem of learning with noise through the lens of the language generation framework proposed by Kleinberg and Mullainathan (Kleinberg & Mullainathan, 2024). In this setting of *language generation in the limit*, there is a public collection $\mathcal{C}$ of infinite languages defined over a countable universe $U$, and an infinite game is played between an adversary and an algorithm. The adversary privately selects a target language $K \in \mathcal{C}$ and reveals the elements of $K$ in an infinite enumeration, where a string $x_t \in K$ is revealed at each time $t$. The algorithm then observes this enumeration and responds with a string $z_t$ at each time step, with the goal that after some *finite* time, every output string is a correct unseen element of $K$. In this model, we can imagine the adversary as providing the training data, and the algorithm as being the LLM which seeks to eventually generate new correct strings. The related problem where the algorithm instead attempts to identify $K$ rather than generate from it was studied by Gold (Gold, 1967) and Angluin (Angluin, 1980a;b). In this line of work, they showed that the identification problem is challenging, with identification being impossible for many collections of languages. In contrast, (Kleinberg & Mullainathan, 2024) surprisingly showed that the generation problem is much more tractable, and in fact every *countable* collection $\mathcal{C}$ can be generated in the limit.

Building on the Kleinberg and Mullainathan model, there has been a large amount of work studying various natural variants of language generation. Li, Raman, and Tewari (Li et al., 2025) introduced notions of uniform and non-uniform generation which quantify the time $t^\star$ after which the algorithm is required to generate unseen strings from the target language. In the original definition of gen-

eration in the limit, the time $t^\star$ can depend on the specific target language and enumeration chosen by the adversary. In contrast, *uniform generation* requires the time $t^\star$ to be independent of both the target language and enumeration, while *non-uniform generation* allows $t^\star$ to depend on the choice of the target language but not the enumeration. (Li et al., 2025) gave precise structural characterizations for collections that are uniformly and non-uniformly generatable. In particular, (Li et al., 2025) and Charikar and Pabbaraju (Charikar & Pabbaraju, 2025a) independently showed that all countable collections are non-uniformly generatable.

Another line of work has focused on strengthening the power of either the adversary or the algorithm. In our paper, we focus on the *noisy* model introduced by Raman and Raman (Raman & Raman, 2025) which allows the adversary to insert extraneous strings from outside the target language into its enumeration. In *noisy generation in the limit*, the adversary is allowed to pick a finite noise level $n^\star \in \mathbb{N}$ and insert $n^\star$ strings from outside the target language to create a noisy enumeration. The algorithm is then given the noisy enumeration (without being told which strings are noise) and must generate in the same fashion as before. (Raman & Raman, 2025) also defined analogous notions of uniform and non-uniform generation in the noisy setting, with two sets of classes based on whether the time $t^\star$ depends on the noise level $n^\star$ picked by the adversary. In uniform and non-uniform *noise-independent* generation, the time $t^\star$ must be independent of the noise level $n^\star$, in addition to the standard requirements for uniform and non-uniform generation. Meanwhile, in the *noise-dependent* settings, the time $t^\star$ can depend on the specific noise level $n^\star$, i.e., $t^\star$ can depend on $n^\star$ in uniform noise-dependent generation, and depend on $n^\star$ and the target language in non-uniform noise-dependent generation. Bai, Panigrahi, and Zhang (Bai et al., 2026) showed that a collection can be noise-*independently* uniformly or non-uniformly generated if and only if the collection can be uniformly or non-uniformly generated when the adversary does not provide any example strings—intuitively this is because the noise-independent property requires the algorithm to generate correctly even when all of the adversary strings are noisy. In contrast, uniform and non-uniform noise-dependent generation is a much more tractable and interesting setting. (Raman & Raman, 2025) showed that all *countable* collections are non-uniformly noise-dependently generatable, but left finding a complete characterization for (uncountable) collections that are non-uniformly noise-dependently generatable as an open question.

A more fine-grained notion of noisy generation was introduced by (Bai et al., 2026), where an algorithm is said to *generate in the limit with noise level $i$* if it can generate correctly when the adversary inserts at most $i$ extraneous strings. They showed a strict hierarchy, where for every noise level $i \geq 0$, there exists a collection $\mathcal{C}_i$ which is generatable in the limit with noise level $i$, but not with noise level $i + 1$.

In this work, we explore an analogous fine-grained notion for uniform and non-uniform generation. We say that a collection $\mathcal{C}$ can be uniformly (resp. non-uniformly) generated with noise level $i$, if there exists an algorithm which uniformly (resp. non-uniformly) generates when the adversary inserts at most $i$ noisy strings. This is a refinement of the noise-*dependent* generatability setting introduced in (Raman & Raman, 2025). Our first result shows a separation between noiseless generation and generation with noise. For both uniform and non-uniform generation, there are collections that can be generated without noise, but cannot be generated with noise level 1. This implies a negative answer to an open question in (Raman & Raman, 2025), which asked if non-uniform generation is equivalent to non-uniform noise-dependent generation. The separation shown here suggests that noisy labels fundamentally affect the generation ability of language models.

We then show a surprising equivalence—for both uniform and non-uniform generation, a collection $\mathcal{C}$ is generatable with noise level $i$ for some $i \geq 1$ if and only if $\mathcal{C}$ is generatable with noise level 1. This result strongly contrasts with the infinite hierarchy for noisy generation in the limit shown by (Bai et al., 2026), and also with the separation we show between noise level 0 and 1 for noisy uniform and non-uniform generation. Finally, we leverage this equivalence to further show that uniform (resp. non-uniform) generation with noise level 1 is equivalent to uniform (resp. non-uniform) noise-dependent generation, and provide the first known characterization for classes that are non-uniformly noise-dependently generatable.

### 1.1. Other Related Work

There has been a plethora of recent work within the language generation in the limit framework. As discussed above, the most relevant line of research in this framework is the work studying noise in language generation, which was initiated by (Raman & Raman, 2025) and extended by (Bai et al., 2026). In contrast to our work which studies finite amounts of noise, Mehrotra, Velegkas, Yu, and Zhou (Mehrotra et al., 2025) investigate the setting of infinite noise, where they show that language generation in the limit is achievable for all countable collections if and only if the fraction of noisy strings converges to zero.

Another line of research has focused on the tradeoff between correctness and breadth of generation. Various notions of breadth have been formalized, which study the "fraction" of strings in the target language that the algorithm eventually generates (Charikar & Pabbaraju, 2025a;

Kalavasis et al., 2025a; Kleinberg & Wei, 2025b;a; Mehrotra et al., 2025; Kalavasis et al., 2025b; Kleinberg & Wei, 2026). A related work by (Peale et al., 2025) investigates "representative" generation, where the target language is divided into groups, and the algorithm is required to generate from all groups. Other work has investigated whether generatability is closed under finite unions (Hanneke et al., 2025; Bai et al., 2026). Within non-uniform generation, (Charikar & Pabbaraju, 2025b) define a notion of pareto-optimality of generation times, and (Arenas et al., 2026) explore bounds on uniform generation times for specific classes of collections. (Kleinberg et al., 2026) introduce the setting of mistake-bounded generation where the metric of an algorithm's success is the number of mistakes instead of the time of the last mistake.

(Anastasopoulos et al., 2026) formalized a notion of safe generation where the algorithm must avoid certain unsafe strings in the target language. (Høgsgaard & Pabbaraju, 2026) study agnostic generation, where the adversary's outputs are not required to be drawn from a particular target language. (Li et al., 2026a) study a model where the universe that strings are drawn from form a metric space. (Racca et al., 2026) define a new model with a replay adversary that is allowed to insert the algorithm's own past outputs into the enumeration. (Mehrotra et al., 2026; Li et al., 2026b) study language generation under the constraint of differential privacy. (Li et al., 2026c) study contrastive generation where the algorithm is presented with contrastive pairs of positive and negative examples. Finally, (Charikar et al., 2025) study list identification where the algorithm produces a list of $k$ language guesses at each time.

## 2. Preliminaries and Results

In this section, we first introduce the Kleinberg-Mullainathan model (Kleinberg & Mullainathan, 2024), and then formally state our results. A language $L$ is an infinite subset of a countably infinite set $U$ called the universe, and a collection $\mathcal{C}$ is a (possibly uncountable) set of languages. Unless specified otherwise, we will assume without loss of generality that all collections are over the set of integers $\mathbb{Z}$. We denote the set of nonnegative integers by $\mathbb{N} = \{0, 1, \dots\}$ and abbreviate contiguous elements of a sequence $x_i, \dots, x_j$ by $x_{i:j}$.

### 2.1. Generation in the Limit

In the general setup, there is a fixed collection $\mathcal{C}$ and a target language $K \in \mathcal{C}$ selected by the adversary. The adversary then presents the strings of $K$ in an enumeration $x_0$, $x_1$, ..., where each $x_t$ is contained in $K$, and for every $z \in K$, there exists some $t$ where $z = x_t$. In addition, we require that every string in the enumeration is unique. We note that in some previous work, the adversary was allowed to

repeat strings in its enumeration. However, it was shown by (Bai et al., 2026) that generation is equivalent when the adversary is not allowed to repeat strings.

At each time step $t$, the algorithm takes as input the ordered list of strings enumerated by the adversary so far and outputs a new string $z_t$. The goal is that after some finite time $t^\star$, all strings $z_t$ for $t \geq t^\star$ are correct *unseen* strings from the target language $K$. Note that unlike the adversary, the algorithm is allowed to output the same string multiple times, but the algorithm's string is only considered correct if it is distinct from all the example strings given by the adversary so far.

For any enumeration $x$, we will use $S(x)_t = \{x_0, x_1, \dots, x_t\}$ to denote the set of strings enumerated up until time $t$. When the enumeration is clear from context, we will simply write $S_t$.

**Definition 2.1** (Generator algorithm). A generator algorithm is a function $U^* \to U$ which takes as input a finite ordered list of strings $x_0, \dots, x_t$, and outputs a string $z_t$.

**Definition 2.2** (Generation in the limit (Kleinberg & Mullainathan, 2024)). An algorithm $G$ generates in the limit for a collection $\mathcal{C}$ if for any $K \in \mathcal{C}$ and any enumeration $x$ of $K$, there exists a time $t^\star$ such that for all $t \geq t^\star$, the generated string $z_t$ at time $t$ is in $K \setminus S_t$.

In the above definition, the time $t^\star$ at which the algorithm must generate correctly can be a function of both the target language $K$ and the adversary's enumeration $x$ of $K$. A stricter requirement would be that $t^\star$ is independent of the enumeration or even the target language. These notions were formalized by (Li et al., 2025) to define uniform and non-uniform generation.

**Definition 2.3** (Uniform generation (Li et al., 2025)). An algorithm $G$ uniformly generates for a collection $\mathcal{C}$ if there exists a time step $t^\star$ such that for any $K \in \mathcal{C}$ and any enumeration $x$ of $K$, the generated string $z_t$ for every time $t \geq t^\star$ is in $K \setminus S_t$.

**Definition 2.4** (Non-uniform generation (Li et al., 2025)). An algorithm $G$ non-uniformly generates for a collection $\mathcal{C}$ if for any $K \in \mathcal{C}$, there exists a time step $t^\star$ such that for every enumeration $x$ of $K$ and every time $t \geq t^\star$, the generated string $z_t$ is in $K \setminus S_t$.

### 2.2. Noisy Generation

Raman and Raman (Raman & Raman, 2025) introduced a model of noisy generation where there may be a finite number of extraneous strings in the adversary's enumeration. As before, the adversary selects a language $K \in \mathcal{C}$ and an enumeration $y_0, y_1, \dots$ of $K$. However the adversary is now allowed to choose a finite noise level $n^\star \in \mathbb{N}$, and insert $n^\star$ *unique* strings not belonging to $K$ into the enumeration $y_0, y_1, \dots$. The resulting *noisy enumeration* $x_0$,

$x_1, \ldots$ is then presented to the algorithm $G$, which must eventually generate correct unseen strings from the target language $K$.

**Definition 2.5** (Enumeration with noise level $i$)**.** For any infinite language $K$ and integer $i \in \mathbb{N}$, an enumeration of $K$ with noise level $i$ is any infinite sequence $x_0$, $x_1$, $\ldots$ without repetitions, such that $K \subseteq \bigcup_{j \in \mathbb{N}} \{x_j\}$ and $\left| \bigcup_{j \in \mathbb{N}} \{x_j\} \setminus K \right| \leq i$.

We have the following definitions for generation with noise based on how the time step $t^\star$ at which we generate correctly is quantified.

**Definition 2.6** (Uniform noise-dependent generation (Raman & Raman, 2025))**.** An algorithm $G$ uniformly noise-dependently generates for a collection $\mathcal{C}$ if for every noise level $n^\star \in \mathbb{N}$, there exists a time $t^\star$ such that for every $K \in \mathcal{C}$, and every enumeration $x$ of $K$ with noise level $n^\star$, the algorithm's output $z_t$ is in $K \setminus S_t$ for all times $t \geq t^\star$.

**Definition 2.7** (Non-uniform noise-dependent generation (Raman & Raman, 2025))**.** An algorithm $G$ non-uniformly noise-dependently generates for a collection $\mathcal{C}$ if for every noise level $n^\star \in \mathbb{N}$ and every $K \in \mathcal{C}$, there exists a time $t^\star$ such that for every enumeration $x$ of $K$ with noise level $n^\star$, the algorithm's output $z_t$ is in $K \setminus S_t$ for all times $t \geq t^\star$.

Note that in the above definitions for uniform and non-uniform noisy generation, the time step $t^\star$ can depend on the noise level. The related notions where $t^\star$ is independent of the noise level are fully characterized by (Raman & Raman, 2025; Bai et al., 2026), and we do not discuss them in this paper.

Building upon notions introduced in (Li et al., 2025), the following useful definitions of consistency and closure were introduced by (Raman & Raman, 2025).

**Definition 2.8** (Consistent languages at noise level $i$ (Raman & Raman, 2025))**.** Given a collection $\mathcal{C}$, the set of consistent languages for a set $S$ at noise level $i$ is the set $\mathcal{C}(S, i) := \{ L \in \mathcal{C} \mid |S \setminus L| \leq i \}$.

**Definition 2.9** (Noisy closure (Raman & Raman, 2025))**.** Given a collection $\mathcal{C}$, the noisy closure of a set $S$ at noise level $i$, denoted by $\langle S \rangle_{\mathcal{C}, i}$, is the intersection of all consistent languages for $S$ in $\mathcal{C}$ at noise level $i$. Formally,

$$\langle S \rangle_{\mathcal{C}, i} := \begin{cases} \bigcap_{L \in \mathcal{C}(S, i)} L & |\mathcal{C}(S, i)| \geq 1 \\ \emptyset & |\mathcal{C}(S, i)| = 0 \end{cases}.$$

Intuitively, the set of consistent languages for a set $S$ at noise level $i$ represents the set of possible target languages given that the adversary has currently revealed the set $S$ in an enumeration with noise level $i$. The noisy closure is then the intersection of those consistent languages, representing

the set of strings which the algorithm can safely generate from.

**Definition 2.10** (Noisy closure dimension (Raman & Raman, 2025))**.** The noisy closure dimension of a collection $\mathcal{C}$ at noise level $i$, denoted $\mathrm{NC}_i(\mathcal{C})$, is the size of the largest finite set $S$ such that $\mathcal{C}(S, i) \neq \emptyset$ and $|\langle S \rangle_{\mathcal{C}, i}| < \infty$. If there are arbitrarily large sets with finite closure, the closure dimension is $\infty$.

The following simple lemma clarifies the role of the noisy closure operator—after seeing any set of example strings $S$, an algorithm generating with noise level $i$ can safely generate from $\langle S \rangle_{\mathcal{C}, i}$.

**Lemma 2.11.** *Let $\mathcal{C}$ be an arbitrary collection and $S$ be an arbitrary subset of the universe. For any noise level $i$ and language $L \in \mathcal{C}(S, i)$, we have $\langle S \rangle_{\mathcal{C}, i} \subseteq L$.*

*Proof.* By definition, $\langle S \rangle_{\mathcal{C}, i} = \bigcap_{L \in \mathcal{C}(S, i)} L$, so $\langle S \rangle_{\mathcal{C}, i} \subseteq L$ for any $L \in \mathcal{C}(S, i)$. $\qquad\square$

Building upon the fine-grained notions of noisy generation in the limit introduced in (Bai et al., 2026), we define the analogous versions for uniform and non-uniform generation.

**Definition 2.12** (Uniform generation with noise level $i$)**.** For any $i \in \mathbb{N}$, an algorithm $G$ uniformly generates with noise level $i$ for a collection $\mathcal{C}$ if there exists a time $t^\star$ such that for every $K \in \mathcal{C}$, every enumeration $x$ of $K$ with noise level at most $i$, and all $t \geq t^\star$, the string generated by the algorithm at time $t$ belongs to $K \setminus S_t$.

**Definition 2.13** (Non-uniform generation with noise level $i$)**.** For any $i \in \mathbb{N}$, an algorithm $G$ non-uniformly generates with noise level $i$ for a collection $\mathcal{C}$ if for every $K \in \mathcal{C}$, there exists a time $t^\star$ such that for every enumeration $x$ of $K$ with noise level at most $i$ and all $t \geq t^\star$, the string generated by the algorithm at time $t$ belongs to $K \setminus S_t$.

### 2.3. Our Results

Our results first focus on understanding the power of uniform and non-uniform generation with finite noise. We then extend our exploration to noise-dependent generation, culminating in a complete landscape for uniform and non-uniform noise-dependent generation. We begin by showing an equivalence in generation between different finite noise levels.

**Theorem 2.14.** *For any finite noise level $i \geq 1$, a collection $\mathcal{C}$ is uniformly generatable with noise level $i$ if and only if $\mathcal{C}$ is uniformly generatable with noise level $1$. Similarly, a collection $\mathcal{C}$ is non-uniformly generatable with noise level $i$ if and only if $\mathcal{C}$ is non-uniformly generatable with noise level $1$.*

The above equivalence is perhaps surprising, since it contrasts with the strict hierarchy for noisy generation in the limit shown by (Bai et al., 2026)—for each noise level $i \geq 0$ there exists a collection that can be noisily generated in the limit with noise level $i$, but not noise level $i + 1$. Meanwhile, our result shows that for uniform and non-uniform noisy generation, the hierarchy collapses for all noise levels $i \geq 1$.

Next we show that for both uniform and non-uniform generation, there is a separation between noiseless generation and generation with noise level 1.

**Theorem 2.15.** *There exists a collection that is uniformly generatable (and hence non-uniformly generatable) without noise, but is not non-uniformly generatable (and hence not uniformly generatable) with noise level* 1.

We note that separating non-uniform generation and non-uniform noise-dependent generation was left as an open question in (Raman & Raman, 2025). Our result negatively answers the question in the strongest possible sense—there is a separation between non-uniform generation and noisy non-uniform generation with just a single noisy string.

We then show that uniform noise-dependent generatability is in fact equivalent to uniform generation with noise level 1. This allows us to derive a simple characterization for collections that are uniformly noise-dependently generatable in terms of the noisy closure dimension. The following theorem summarizes our results, which precisely characterize the landscape of uniform noise-dependent generation.

**Theorem 2.16.** *For any collection $\mathcal{C}$, the following are equivalent:*

1. *$\mathrm{NC}_1(\mathcal{C}) < \infty$,*

2. *$\mathcal{C}$ is uniformly generatable with noise level $i$, for any $i \geq 1$,*

3. *$\mathcal{C}$ is uniformly noise-dependently generatable.*

*However, there exists a collection that is uniformly generatable without noise, but is not uniformly generatable with noise level* 1.

Our characterization for noise-dependent generatability, that $\mathrm{NC}_1(\mathcal{C}) < \infty$, simplifies the existing characterization given in (Raman & Raman, 2025), which states that a collection $\mathcal{C}$ is uniformly noise-dependently generatable if and only if $\mathrm{NC}_i(\mathcal{C}) < \infty$ for every $i \in \mathbb{N}$.

Finally, we take the same approach for non-uniform noise-dependent generation and show a similar equivalence.

**Theorem 2.17.** *For any collection $\mathcal{C}$, the following are equivalent:*

1. *there exists a countable sequence of collections $\mathcal{C}_0 \subseteq \mathcal{C}_1 \subseteq \ldots$ such that $\mathcal{C} = \bigcup_{j \in \mathbb{N}} \mathcal{C}_j$ and $\mathrm{NC}_1(\mathcal{C}_j) < \infty$ for all $j \in \mathbb{N}$.*

2. *$\mathcal{C}$ is non-uniformly generatable with noise level $i$, for any $i \geq 1$,*

3. *$\mathcal{C}$ is non-uniformly noise-dependently generatable.*

*However, there exists a collection that is non-uniformly generatable without noise, but is not non-uniformly generatable with noise level* 1.

Our result provides the first known characterization for non-uniform noise-dependent generation, answering another open question from (Raman & Raman, 2025).

## 3. Noisy uniform generation

In this section, we prove the portion of our results concerning noisy uniform generation. We first give an initial complete characterization for collections that are uniformly generatable with noise level $i$. We then use this characterization to show that uniform generation with noise level $i$ is in fact equivalent for all finite $i \geq 1$.

The following lemma uses standard techniques, and is implicitly contained in Theorem 3.3 from (Raman & Raman, 2025). The proof is deferred to Appendix A.

**Lemma 3.1.** *A collection $\mathcal{C}$ is uniformly generatable with noise level $i$ if and only if $\mathrm{NC}_i(\mathcal{C}) < \infty$.*

We now begin to show that for any $i \geq 1$, uniform generation with noise level $i$ is equivalent to uniform generation with noise level 1. The next lemma is the crux of the argument.

**Lemma 3.2.** *For any collection $\mathcal{C}$ and noise level $i \geq 2$, we have $\mathrm{NC}_{i-1}(\mathcal{C}) \geq \left\lfloor \sqrt{\mathrm{NC}_i(\mathcal{C})} \right\rfloor$. In particular, if $\mathrm{NC}_i(\mathcal{C}) = \infty$, then $\mathrm{NC}_{i-1}(\mathcal{C}) = \infty$.*

*Proof.* Assume that $\mathrm{NC}_i(\mathcal{C}) \geq k^2$ for some integer $k \geq 1$. We will show that $\mathrm{NC}_{i-1}(\mathcal{C}) \geq k$. Since $\mathrm{NC}_i(\mathcal{C}) \geq k^2$, there exists some set $S$ of size $k^2$ such that $\mathcal{C}(S, i) \neq \emptyset$ and $|\langle S \rangle_{\mathcal{C}, i}| < \infty$. Without loss of generality, we can assume that $S = \{1, \ldots, k^2\}$. We first wish to construct a partition $S_1 \sqcup \cdots \sqcup S_k$ of $S$, where each $S_j$ has size exactly $k$, and for every $S_j$ there is at least one language in $\mathcal{C}$ that is consistent with $S_j$ at noise level $i - 1$. If $k \leq i - 1$, we can partition $S$ into any $k$ equal-sized sets $S_1, S_2, \ldots, S_k$. For any $S_j$, every language in $\mathcal{C}$ is consistent with $S_j$ at noise level $|S_j| = k$. Since $k \leq i - 1$, each language is also consistent with $S_j$ at noise level $i - 1$, so we have $\mathcal{C}(S_j, i - 1) = \mathcal{C}$ and $\mathcal{C}(S_j, i - 1) \neq \emptyset$ for all $S_j$.

Now consider when $k \geq i$. If there exists some language $L_0 \in \mathcal{C}(S, i)$ that is also consistent with $S$ at noise level

$i - 1$, then for any partition of $S$ into $k$ equal-sized sets $S_1$, $S_2$, ..., $S_k$, we have $L_0 \in \mathcal{C}(S_j, i - 1)$ for each $S_j$. Otherwise, there does not exist a language in $\mathcal{C}(S, i)$ that is consistent with $S$ at noise level $i - 1$, so $|S \setminus L| = i$ for all languages $L \in \mathcal{C}(S, i)$. That is, every language in $\mathcal{C}(S, i)$ is missing exactly $i$ elements from $S$. Let $L_1 \in \mathcal{C}(S, i)$ be any language, and let $S \setminus L_1 = \{y_1, y_2, \ldots, y_i\}$ be the $i$ elements that $L_1$ is missing from $S$. Since $k \geq i$, we can partition $S$ into $k$ equal-sized sets $S_1$, $S_2$, ..., $S_k$, such that each $y_j$ lies in a distinct set of the partition. Each $S_j$ contains at most one element from $S \setminus L_1$, so $L_1 \in \mathcal{C}(S_j, 1)$ for all $S_j$. Since $i - 1 \geq 1$, this implies $L_1 \in \mathcal{C}(S_j, i - 1)$ for all $S_j$ as desired.

In all cases, we have now constructed the desired partition $S_1 \sqcup \cdots \sqcup S_k$ of $S$, where each $S_j$ has size exactly $k$, and for every $S_j$ there is at least one language in $\mathcal{C}$ that is consistent with $S_j$ at noise level $i - 1$.

If there is some $S_j$ where $\mathcal{C}(S_j, i - 1) \neq \emptyset$ and $|\langle S_j \rangle_{\mathcal{C}, i-1}| < \infty$, then we have by definition that $\mathrm{NC}_{i-1}(\mathcal{C}) \geq k$. Otherwise, since we constructed each $S_j$ to satisfy $\mathcal{C}(S_j, i - 1) \neq \emptyset$, we are in the case where $|\langle S_j \rangle_{\mathcal{C}, i-1}| = \infty$ for every $S_j$. Roughly speaking, we will construct a set $A$ of size $k$ that contains one element from the closure of each $S_j$ at noise level $i - 1$. Then, we show that any language $L$ that is consistent with $S$ at noise level $i$ must be consistent with at least $k - 1$ of the sets $S_j$ at noise level $i - 1$. Since the elements of $A$ are constructed from the closures of the sets $S_j$ at noise level $i - 1$, this implies that each such language $L$ contains at least $k - 1$ of the elements in $A$, so each such $L$ must be consistent with $A$ at noise level $1$. Thus, the closure of $A$ at noise level $1$ is a subset of the closure of $S$ at noise level $i$, implying that the closure of $A$ at noise level $i - 1 \geq 1$ is finite, so $\mathrm{NC}_{i-1}(\mathcal{C}) \geq |A| = k$ as desired.

More formally, we iteratively build a set $A$ of size $k$ which has finite closure at noise level $i - 1$ in $k$ iterations. At iteration $j$, take $x_j$ to be an arbitrary element from $\langle S_j \rangle_{\mathcal{C}, i-1} \setminus \{x_1, \ldots, x_{j-1}\}$. Since $|\langle S_j \rangle_{\mathcal{C}, i-1}| = \infty$, such an element must always exist. Then, we simply let $A = \{x_1, \ldots, x_k\}$. We wish to show that $\langle A \rangle_{\mathcal{C}, i-1} \subseteq \langle S \rangle_{\mathcal{C}, i}$, since that would imply $|\langle A \rangle_{\mathcal{C}, i-1}| \leq |\langle S \rangle_{\mathcal{C}, i}| < \infty$, which would imply $\mathrm{NC}_{i-1}(\mathcal{C}) \geq |A| = k$ as desired. To show $\langle A \rangle_{\mathcal{C}, i-1} \subseteq \langle S \rangle_{\mathcal{C}, i}$, we prove that $\mathcal{C}(S, i) \subseteq \mathcal{C}(A, 1)$, i.e., every language consistent with $S$ at noise level $i$ must be consistent with $A$ at noise level $1$. Let $L \in \mathcal{C}(S, i)$ be an arbitrary language that is consistent with $S$ at noise level $i$. We claim that there is at most one $j$ such that $x_j \notin L$.

Assume for contradiction that there are two indices $\ell \neq \ell'$ such that $x_\ell \notin L$ and $x_{\ell'} \notin L$. Since each $x_j \in A$ is drawn from $\langle S_j \rangle_{\mathcal{C}, i-1}$, we have by Lemma 2.11 that $x_j \in L'$ for each $L' \in \mathcal{C}(S_j, i - 1)$. Because $x_l \notin L$ and $x_{\ell'} \notin L$, it must be that $L \notin \mathcal{C}(S_\ell, i - 1)$ and $L \notin \mathcal{C}(S_{\ell'}, i - 1)$. Then

by definition, we have $|S_\ell \setminus L| \geq i$ and $|S_{\ell'} \setminus L| \geq i$. Since $S_\ell$ and $S_{\ell'}$ are disjoint, this means that $|(S_\ell \cup S_{\ell'}) \setminus L| \geq 2i$. However, this would imply that $|S \setminus L| \geq 2i$, which is a contradiction because $L$ is consistent with $S$ at noise level $i$.

Thus, there can be at most one $j$ such that $x_j \notin L$, implying that $L$ is consistent with $A$ at noise level $1$. Because $L$ was arbitrary, we have $\mathcal{C}(S, i) \subseteq \mathcal{C}(A, 1)$, which also gives $\mathcal{C}(S, i) \subseteq \mathcal{C}(A, i - 1)$ because $i - 1 \geq 1$. This implies that $\langle A \rangle_{\mathcal{C}, i-1} \subseteq \langle S \rangle_{\mathcal{C}, i}$. Since $|\langle S \rangle_{\mathcal{C}, i}| < \infty$, we also have $|\langle A \rangle_{\mathcal{C}, i-1}| < \infty$. Finally, because $|A| = k$, we have $\mathrm{NC}_{i-1}(\mathcal{C}) \geq k$. This implies $\mathrm{NC}_{i-1}(\mathcal{C}) \geq \left\lfloor \sqrt{\mathrm{NC}_i(\mathcal{C})} \right\rfloor$ as desired.

Note that if $\mathrm{NC}_i(\mathcal{C}) = \infty$, then $\mathrm{NC}_i(\mathcal{C}) \geq k^2$ for every $k \in \mathbb{N}$. Thus, $\mathrm{NC}_{i-1}(\mathcal{C}) \geq k$ for every $k \in \mathbb{N}$, so $\mathrm{NC}_{i-1}(\mathcal{C}) = \infty$. $\qquad\square$

We now prove the first part of Theorem 2.14.

**Theorem 3.3.** *For any collection $\mathcal{C}$ and finite noise level $i \geq 1$, the collection $\mathcal{C}$ is uniformly generatable with noise level $i$ if and only if $\mathcal{C}$ is uniformly generatable with noise level $1$.*

*Proof.* Clearly if a collection $\mathcal{C}$ is uniformly generatable with noise level $i$ for some $i \geq 1$, then $\mathcal{C}$ is also uniformly generatable with noise level $1$. For the other direction, we have by Lemma 3.1 that $\mathcal{C}$ is uniformly generatable with noise level $1$ if and only if $\mathrm{NC}_1(\mathcal{C}) < \infty$. From the contrapositive of Lemma 3.2, we have for any $i \geq 2$ that $\mathrm{NC}_{i-1}(\mathcal{C}) < \infty$ implies $\mathrm{NC}_i(\mathcal{C}) < \infty$. Thus if $\mathrm{NC}_1(\mathcal{C}) < \infty$, then $\mathrm{NC}_i(\mathcal{C}) < \infty$ for every $i \geq 1$. This implies that $\mathcal{C}$ is uniformly generatable with noise level $i$ for every $i \geq 1$. $\qquad\square$

Our results naturally yield a simpler characterization for uniform noise-dependent generation. We first state the following characterization that was proved in (Raman & Raman, 2025).

**Theorem 3.4** (Theorem 3.3 in (Raman & Raman, 2025))**.** *A collection $\mathcal{C}$ is uniformly noise-dependently generatable if and only if $\mathrm{NC}_i(\mathcal{C}) < \infty$ for all $i \geq 1$.*

Since uniform generation with noise level $i$ is equivalent for all $i \geq 1$, we obtain the following characterization.

**Theorem 3.5.** *A collection $\mathcal{C}$ is uniformly noise-dependently generatable if and only if $\mathrm{NC}_1(\mathcal{C}) < \infty$.*

*Proof.* From Theorem 3.3 and Lemma 3.1, we have for any $i \geq 1$ that $\mathrm{NC}_i(\mathcal{C}) < \infty$ if and only if $\mathrm{NC}_1(\mathcal{C}) < \infty$. Applying Theorem 3.4, we have the desired result. $\qquad\square$

In summary, we show that uniform generation with noise level 1 is equivalent to uniform generation with noise level $i$ for any $i \geq 1$, which is further equivalent to uniform noise-dependent generation. These forms of generation are all characterized by collections $\mathcal{C}$ that satisfy $\mathrm{NC}_1(\mathcal{C}) < \infty$.

**Theorem 2.16.** *For any collection $\mathcal{C}$, the following are equivalent:*

1. *$\mathrm{NC}_1(\mathcal{C}) < \infty$,*

2. *$\mathcal{C}$ is uniformly generatable with noise level $i$, for any $i \geq 1$,*

3. *$\mathcal{C}$ is uniformly noise-dependently generatable.*

*However, there exists a collection that is uniformly generatable without noise, but is not uniformly generatable with noise level 1.*

*Proof.* The equivalence of Item 1 and Item 2 is given by Theorem 3.3 and Lemma 3.1. The equivalence of Item 1 and Item 3 is given by Theorem 3.5. To see that there exists a collection that is uniformly generatable without noise, but is not uniformly generatable with noise level 1, we note that (Raman & Raman, 2025) (Theorem 3.5) proved there exists a collection $\mathcal{C}$ which is uniformly generatable without noise, but is not uniformly noise-dependently generatable. Since uniform noise-dependent generatability is equivalent to uniform generatability with noise level 1, this gives a collection that is uniformly generatable without noise, but is not uniformly generatable with noise level 1. $\square$

## 4. Noisy non-uniform generation

In this section, we prove the portion of our results concerning noisy non-uniform generation. We first give an initial complete characterization for collections that are non-uniformly generatable with noise level $i$. Then, we leverage the results shown in the previous section to prove the corresponding equivalences for noisy non-uniform generation.

The following lemma uses standard techniques, and the proof is deferred to Appendix A.

**Lemma 4.1.** *A collection $\mathcal{C}$ is non-uniformly generatable with noise level $i$ if and only if there exists a countable sequence of collections $\mathcal{C}_0 \subseteq \mathcal{C}_1 \subseteq \ldots$ such that $\mathcal{C} = \bigcup_{j \in \mathbb{N}} \mathcal{C}_j$ and $\mathrm{NC}_i(\mathcal{C}_j) < \infty$ for all $j \in \mathbb{N}$.*

We now prove the second half of Theorem 2.14.

**Theorem 4.2.** *For any collection $\mathcal{C}$ and finite noise level $i \geq 1$, the collection $\mathcal{C}$ is non-uniformly generatable with noise level $i$ if and only if $\mathcal{C}$ is non-uniformly generatable with noise level 1.*

*Proof.* Clearly if $\mathcal{C}$ is non-uniformly generatable with noise level $i$ for some $i \geq 1$, then $\mathcal{C}$ is non-uniformly generatable with noise level 1. For the other direction, assume that $\mathcal{C}$ is non-uniformly generatable with noise level 1. From Lemma 4.1, there must exist a countable sequence of collections $\mathcal{C}_0 \subseteq \mathcal{C}_1 \subseteq \ldots$ such that $\mathcal{C} = \bigcup_{j \in \mathbb{N}} \mathcal{C}_j$ and $\mathrm{NC}_1(\mathcal{C}_j) < \infty$ for all $j \in \mathbb{N}$. Applying Theorem 3.3 and Lemma 3.1, we have for any noise level $i \geq 1$ that $\mathrm{NC}_i(\mathcal{C}_j) < \infty$ for every $j \in \mathbb{N}$. Thus by Lemma 4.1, $\mathcal{C}$ is non-uniformly generatable with noise level $i$ as desired. $\square$

We now provide a complete characterization for non-uniform noise-dependent generation. We first state the sufficient and necessary conditions for non-uniform noise-dependent generation proved in (Raman & Raman, 2025).

**Lemma 4.3** (Lemma 3.6 in (Raman & Raman, 2025))**.** *A collection $\mathcal{C}$ is non-uniformly noise-dependently generatable if there exists a countable sequence of collections $\mathcal{C}_0 \subseteq \mathcal{C}_1 \subseteq \ldots$ such that $\mathcal{C} = \bigcup_{i \in \mathbb{N}} \mathcal{C}_i$ and $\mathrm{NC}_i(\mathcal{C}_i) < \infty$ for all $i \in \mathbb{N}$.*

**Lemma 4.4** (Lemma 3.8 in (Raman & Raman, 2025))**.** *If a collection $\mathcal{C}$ is non-uniformly noise-dependently generatable, then for every noise level $i \geq 1$, there exists a countable sequence of collections $\mathcal{C}_0 \subseteq \mathcal{C}_1 \subseteq \ldots$ such that $\mathcal{C} = \bigcup_{j \in \mathbb{N}} \mathcal{C}_j$ and $\mathrm{NC}_i(\mathcal{C}_j) < \infty$ for all $j \in \mathbb{N}$.*

It is not immediately clear whether the two conditions above are equivalent, and in fact (Raman & Raman, 2025) conjectured that the true characterization for non-uniform noise-dependent generatability would need to go beyond the sufficient and necessary conditions given above. However, we show surprisingly that the conditions in Lemma 4.3 and Lemma 4.4 are in fact both equivalent to non-uniform generation with noise level 1. Thus, non-uniform noise-dependent generatability is also equivalent to non-uniform generation with noise level 1.

**Lemma 4.5.** *A collection $\mathcal{C}$ is non-uniformly noise-dependently generatable if there exists a countable sequence of collections $\mathcal{C}_0 \subseteq \mathcal{C}_1 \subseteq \ldots$ such that $\mathcal{C} = \bigcup_{i \in \mathbb{N}} \mathcal{C}_i$ and $\mathrm{NC}_1(\mathcal{C}_i) < \infty$ for all $i \in \mathbb{N}$.*

*Proof.* Let $\mathcal{C}_0 \subseteq \mathcal{C}_1 \subseteq \ldots$ be a countable sequence of collections such that $\mathcal{C} = \bigcup_{i \in \mathbb{N}} \mathcal{C}_i$ and $\mathrm{NC}_1(\mathcal{C}_i) < \infty$ for all $i \in \mathbb{N}$. By Lemma 4.3, it suffices to show that there exists a countable sequence of collections $\mathcal{C}_0 \subseteq \mathcal{C}_1 \subseteq \ldots$ such that $\mathcal{C} = \bigcup_{i \in \mathbb{N}} \mathcal{C}_i$ and $\mathrm{NC}_i(\mathcal{C}_i) < \infty$ for all $i \in \mathbb{N}$. This is true because by Theorem 3.3 and Lemma 3.1, $\mathrm{NC}_1(\mathcal{C}_i) < \infty$ implies $\mathrm{NC}_i(\mathcal{C}_i) < \infty$ for any $i \in \mathbb{N}$. $\square$

**Lemma 4.6.** *If a collection $\mathcal{C}$ is non-uniformly noise-dependently generatable then there exists a countable sequence of collections $\mathcal{C}_0 \subseteq \mathcal{C}_1 \subseteq \ldots$ such that $\mathcal{C} = \bigcup_{i \in \mathbb{N}} \mathcal{C}_i$ and $\mathrm{NC}_1(\mathcal{C}_i) < \infty$ for all $i \in \mathbb{N}$.*

*Proof.* Let $\mathcal{C}$ be a collection that is non-uniformly noise-dependently generatable. By Lemma 4.4, for every noise level $i \geq 1$, there exists a countable sequence of collections $\mathcal{C}_0 \subseteq \mathcal{C}_1 \subseteq \ldots$ such that $\mathcal{C} = \bigcup_{j \in \mathbb{N}} \mathcal{C}_j$ and $\mathrm{NC}_i(\mathcal{C}_j) < \infty$ for all $j \in \mathbb{N}$. The desired statement is the case when $i = 1$. $\qquad\square$

Combining Lemmas 4.5 and 4.6, we have the following complete characterization for non-uniform noise-dependent generatability.

**Theorem 4.7.** *A collection $\mathcal{C}$ is non-uniformly noise-dependently generatable if and only if there exists a countable sequence of collections $\mathcal{C}_0 \subseteq \mathcal{C}_1 \subseteq \ldots$ such that $\mathcal{C} = \bigcup_{i \in \mathbb{N}} \mathcal{C}_i$ and $\mathrm{NC}_1(\mathcal{C}_i) < \infty$ for all $i \in \mathbb{N}$.*

This gives the first known characterization for non-uniform noise-dependent generatability, answering an open question from (Raman & Raman, 2025).

We now turn to separating non-uniform generation with and without noise. Unlike uniform noisy generation, it is an open question whether non-uniform generation is equivalent to non-uniform noise-dependent generation. We resolve this question by showing that there is a collection that is uniformly generatable without noise, but is not non-uniformly generatable with noise level 1.

**Theorem 2.15.** *There exists a collection that is uniformly generatable (and hence non-uniformly generatable) without noise, but is not non-uniformly generatable (and hence not uniformly generatable) with noise level 1.*

*Proof.* Let the universe $U$ be $\mathbb{N} \times \mathbb{N}$ and let $B_c = \{(c,i) \mid i \in \mathbb{N}\}$ for each $c \in \mathbb{N}$. For each nonempty subset $x \subseteq \mathbb{N}$, define the language $L_x = \bigcup_{c \in x} B_c$. Finally, let $\mathcal{C} = \{L_x \mid x \subseteq \mathbb{N}, x \neq \emptyset\}$ be the collection of all such languages. Intuitively, we can imagine each of the sets $B_c$ as being disjoint infinite "columns". Then, our collection $\mathcal{C}$ consists exactly of all nonempty unions of those columns. For any pair $x = (a, b) \in \mathbb{N} \times \mathbb{N}$, we write $f(x) \coloneqq a$ to denote the first coordinate of the pair $x$.

We first show that $\mathcal{C}$ is uniformly generatable without noise. Let $c = f(x_0)$ be the column of the first adversary string. Note that this implies $B_c$ is contained in the target language $K$. At each time $t$, the algorithm outputs an arbitrary string from $B_c \setminus S_t$. Since we know that $B_c \subseteq K$, every string output by the algorithm will be correct.

We now claim that $\mathcal{C}$ is not non-uniformly generatable with noise level 1. First define an infinite sequence of sets

$$s_1 = \{0\}, s_2 = \{1, 2\}, s_3 = \{3, 4, 5\}, \ldots,$$

so that each $s_i$ is a set of $i$ elements which is disjoint from every other $s_j$. Correspondingly, define the following infinite sequence of ordered lists of elements of $U$:

$$s_1' = ((0,0)), s_2' = ((1,0),(2,0)),$$
$$s_3' = ((3,0),(4,0),(5,0)), \ldots.$$

We can think of each $s_i$ as being a set of column indices, while the corresponding $s_i'$ is an ordered list containing the first element of each column in $s_i$.

Assume for contradiction that there is some algorithm $G$ which generates non-uniformly with noise level 1 for $\mathcal{C}$. There must either be an infinite number of indices $i$ such that $f(G(s_i')) \in s_i$, or there must be an infinite number of indices $i$ such that $f(G(s_i')) \notin s_i$. We first analyze the case where there are an infinite number of indices $i$ such that $f(G(s_i')) \in s_i$. Let $X = \{i \mid f(G(s_i')) \in s_i\}$. Now consider the language

$$L = \bigcup_{i \in X} \left( \bigcup_{c \in s_i \setminus \{f(G(s_i'))\}} B_c \right).$$

Intuitively, $L$ is the union of columns in $(s_i \setminus \{f(G(s_i'))\})$ for each $s_i$ where $f(G(s_i')) \in s_i$. Note that for each $i \in X$, the list $s_i'$ is a valid prefix of an enumeration of $L$ with noise level 1. Furthermore, we constructed $L$ so that the column $f(G(s_i'))$ is not in $L$, implying that the output $G(s_i')$ is incorrect for each such $i$. Since $X$ is infinite, there are arbitrarily long lists $s_i'$ where $G(s_i')$ is incorrect. Thus, there is no time step where $G$ generates correctly for $L$, so $G$ does not generate non-uniformly with noise level 1.

We now analyze the other case where there are infinitely many indices $i_0 < i_1 < \cdots$ such that $f(G(s_{i_j}')) \notin s_{i_j}$ for all $j \in \mathbb{N}$. For convenience, let $a_j = s_{i_j}$ and $a_j' = s_{i_j}'$. Suppose first there is some set $a_i$ where there are an infinite number of lists $a_j'$ such that $f(G(a_j')) \in a_i$. In this case, let $Y = \{j \mid f(G(a_j')) \in a_i\} \setminus \{i\}$ and consider the language

$$L = \bigcup_{j \in Y} \left( \bigcup_{c \in a_j} B_c \right).$$

Since the columns of $a_i$ are not contained in $L$, and $a_i$ is disjoint from each $a_j$, it must be that $G(a_j') \notin L$ for each $j \in Y$. As before, $Y$ is infinite, so $G$ does not generate non-uniformly with noise level 1.

We are now left with the case where for each set $a_i$, there are only a finite number of lists $a_j'$ such that $f(G(a_j')) \in a_i$. Informally, we iteratively construct a language $L$ by going through the sets $a_i$ in order, and adding a set of columns $a_i$ to $L$ if $a_i$ does not contain any elements of $G(a_j')$ for any previously added set $a_j$, and $G(a_i')$ is not already in $L$.

Consider the sets $L$, $C$, and $N$ constructed by Algorithm 1. Interpreting $C$ as a set of column indices, it is easy to see

**Algorithm 1** Constructing a language $L$

1: $L = \emptyset$
2: $C = \emptyset$
3: $N = \emptyset$
4: **for** $i \in 0, 1, \ldots$ **do**
5:    **if** $a_i \cap N = \emptyset$ and $G(a_i') \notin L$ **then**
6:       $L = L \cup \left( \bigcup_{c \in a_i} B_c \right)$
7:       $C = C \cup \{i\}$
8:       $N = N \cup \{f(G(a_i'))\}$
9:    **end if**
10: **end for**

that $L$ is the union of the columns in $C$. We first argue that $C$ must be infinite. We wish to show that at every iteration $i$ of the for loop on line 4, there is some $j \geq i$ such that $a_j \cap N = \emptyset$ and $G(a_j') \notin L$. Let $i$ be an arbitrary iteration, and let $L_i$, $C_i$, and $N_i$ be the current values of the sets at this iteration. First note that $N_i$ must be finite because the size of $N$ increases by at most 1 in each iteration. Thus, there are only a finite number of indices $j \geq i$ where $a_j \cap N_i \neq \emptyset$. We also have that for each set $a_i$, there are only a finite number of lists $a_j'$ such that $f(G(a_j')) \in a_i$. Since the set $L_i$ corresponds to a finite number of columns, there must only be a finite number of lists $a_j'$ such that $G(a_j') \in L$. Thus in total, there are only a finite number of indices $j \geq i$ where $a_j \cap N_i \neq \emptyset$ or $G(a_j') \in L$, implying that there exists some $j \geq i$ where the if condition on line 5 is true.

We have shown that at each iteration $i$ on line 4, there exists a future iteration $j$ where the size of $C$ increases. Thus $C$ must be an infinite set. We now claim that for each index $i \in C$, we have $G(a_i') \notin L$. By the condition on line 5, we must have had $G(a_i') \notin L$ at iteration $i$ when $i$ was added to $C$. Then at any future iteration $j \geq i$, we have $f(G(a_i')) \in N$. If a column $a_j$ is added to $L$, it must be that $a_j \cap N = \emptyset$, implying that $f(G(a_i')) \notin a_j$. This ensures $G(a_i') \notin L$ as desired. Finally, note that $a_i'$ is a valid prefix of an enumeration of $L$ for each $i \in C$. Since $C$ is infinite, there are arbitrarily large lists $a_i'$ such that $G(a_i') \notin L$, implying $G$ does not generate non-uniformly with noise level 1. $\square$

In summary, we have the following complete picture for noisy non-uniform generation.

**Theorem 2.17.** *For any collection $\mathcal{C}$, the following are equivalent:*

1. *there exists a countable sequence of collections $\mathcal{C}_0 \subseteq \mathcal{C}_1 \subseteq \ldots$ such that $\mathcal{C} = \bigcup_{j \in \mathbb{N}} \mathcal{C}_j$ and $\mathrm{NC}_1(\mathcal{C}_j) < \infty$ for all $j \in \mathbb{N}$.*

2. *$\mathcal{C}$ is non-uniformly generatable with noise level $i$, for any $i \geq 1$,*

3. *$\mathcal{C}$ is non-uniformly noise-dependently generatable.*

*However, there exists a collection that is non-uniformly generatable without noise, but is not non-uniformly generatable with noise level 1.*

*Proof.* The equivalence of Item 1 and Item 2 is given by Theorem 4.2 and Lemma 4.1. The equivalence of Item 1 and Item 3 is given by Theorem 4.7. The last part of the statement is given by Theorem 2.15. $\square$

## 5. Closing Remarks

Language generation in the limit is an exciting framework that captures the fundamental problem of generating new examples based on training data. In this paper, we further explore and quantify the fine-grained influence of noise within this model. We show a sharp separation between noiseless and noisy generation, and also show a surprising equivalence between many models of noisy generation, resolving several open questions from (Raman & Raman, 2025). We hope that the results and techniques in our work will spur future exploration in this area.

## Acknowledgements

We would like to thank Debmalya Panigrahi for many helpful discussions and suggestions in the writing of this paper.

## Impact Statement

This paper presents work whose goal is to advance the field of machine learning. There are many potential societal consequences of our work, none of which we feel must be specifically highlighted here.

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

# A. Omitted proofs

*Proof of Lemma 3.1.* For one direction, assume that $\mathrm{NC}_i(\mathcal{C}) < \infty$. We construct an algorithm $G$ where all outputs after time $\mathrm{NC}_i(\mathcal{C})$ are correct. At any time $t$, if $\langle S_t \rangle_{\mathcal{C},i} \setminus S_t \neq \emptyset$, then $G$ simply outputs an arbitrary element from $\langle S_t \rangle_{\mathcal{C},i} \setminus S_t$. Otherwise, $G$ outputs an arbitrary element. Note that when $|S_t| > \mathrm{NC}_i(\mathcal{C})$, we have by definition that $|\langle S_t \rangle_{\mathcal{C},i}| = \infty$. Thus for all times $t > \mathrm{NC}_i(\mathcal{C})$, the algorithm successfully outputs an element from $\langle S_t \rangle_{\mathcal{C},i} \setminus S_t$. By Lemma 2.11, $\langle S_t \rangle_{\mathcal{C},i}$ is always a subset of the target language, so the algorithm's output at all times $t > \mathrm{NC}_i(\mathcal{C})$ must be a correct unseen string.

For the other direction, assume that $\mathrm{NC}_i(\mathcal{C}) = \infty$ and assume for contradiction that there exists an algorithm $G$ which uniformly generates with noise level $i$ for $\mathcal{C}$. Let $t^\star$ be the time at which $G$ generates correct unseen strings for all $t \geq t^\star$. Since $\mathrm{NC}_i(\mathcal{C}) = \infty$, there must exist a set $S$ with size $|S| \geq t^\star$ such that $|\langle S \rangle_{\mathcal{C},i}| < \infty$. Let $S' = S \cup \langle S \rangle_{\mathcal{C},i}$. We first claim that $\mathcal{C}(S,i) = \mathcal{C}(S',i)$. Since $S \subseteq S'$, it must be that $\mathcal{C}(S',i) \subseteq \mathcal{C}(S,i)$. To see that $\mathcal{C}(S,i) \subseteq \mathcal{C}(S',i)$, let $L \in \mathcal{C}(S,i)$ be an arbitrary language consistent with $S$ at noise level $i$. By Lemma 2.11, we have $(S' \setminus S) \subseteq L$. Thus $|S' \setminus L| = |S \setminus L| \leq i$, so $L \in \mathcal{C}(S',i)$ as desired.

Because $\mathcal{C}(S,i) = \mathcal{C}(S',i)$, we have $\langle S' \rangle_{\mathcal{C},i} = \langle S \rangle_{\mathcal{C},i} \subseteq S'$. Let $t' = |S'|$ and arbitrarily enumerate $S' = \{x_1, \ldots, x_{t'}\}$. Now consider the algorithm's output $z_{t'} = G(x_{1:t'})$. Since $t' \geq t^\star$, the output $z_{t'}$ should be an unseen string from the target language. However, $\langle S' \rangle_{\mathcal{C},i} \subseteq S'$, so for any $z_{t'} \notin S'$, there must exist some $K \in \mathcal{C}(S',i)$ such that $z_{t'} \notin K$. For such a $K \in \mathcal{C}$, the sequence $x_1, \ldots, x_{t'}$ is a valid prefix of an enumeration of $K$ with noise level $i$, and the output $z_{t'}$ is not in $K$. Thus $G$ does not generate uniformly with noise level $i$. □

*Proof of Lemma 4.1.* Assume there exists an algorithm $G$ that non-uniformly generates with noise level $i$ for $\mathcal{C}$. We show that there exists a countable sequence of collections $\mathcal{C}_0 \subseteq \mathcal{C}_1 \subseteq \ldots$ with the desired properties. For each language $L \in \mathcal{C}$, let $t^\star(L)$ be the time at which $G$ must generate correctly for $L$. Since $G$ non-uniformly generates with noise level $i$ for $\mathcal{C}$, each language $L$ must have a finite time $t^\star(L)$. For each $j \in \mathbb{N}$, let $\mathcal{C}_j = \{L \mid t^\star(L) \leq j\}$. It is clear that this forms a sequence of collections $\mathcal{C}_0 \subseteq \mathcal{C}_1 \subseteq \ldots$. Since each language $L \in \mathcal{C}$ has a finite time $t^\star(L)$, we have $\mathcal{C} = \bigcup_{j \in \mathbb{N}} \mathcal{C}_j$. Finally, to see that $\mathrm{NC}_i(\mathcal{C}_j) < \infty$ for all $j \in \mathbb{N}$, note that for every $\mathcal{C}_j$, we have $t^\star(L) < \infty$ for all $L \in \mathcal{C}_j$. Thus, $G$ uniformly generates with noise level $i$ for each $\mathcal{C}_j$, so we have by Lemma 3.1 that $\mathrm{NC}_i(\mathcal{C}_j) < \infty$ for all $j \in \mathbb{N}$.

For the other direction, assume there exists a countable sequence of collections $\mathcal{C}_0 \subseteq \mathcal{C}_1 \subseteq \ldots$ such that $\mathcal{C} = \bigcup_{j \in \mathbb{N}} \mathcal{C}_j$ and $\mathrm{NC}_i(\mathcal{C}_j) < \infty$ for all $j \in \mathbb{N}$. By Lemma 3.1, for any $j \in \mathbb{N}$, there must exist an algorithm $G_j$ which uniformly generates with noise level $i$ for the collection $\mathcal{C}_j$. For each $\mathcal{C}_j$, let $t^\star(\mathcal{C}_j)$ be the time at which algorithm $G_j$ generates correctly for $\mathcal{C}_j$. We design an algorithm $G$ that non-uniformly generates with noise level $i$ for $\mathcal{C}$. At any time $t$, let index $j_t = \max(\{0\} \cup \{j \leq t \mid t^\star(\mathcal{C}_j) \leq t\})$. Then our algorithm simply outputs according to the algorithm $G_{j_t}$, i.e., $G(x_0, \ldots, x_t) = G_{j_t}(x_0, \ldots, x_t)$. Intuitively, our algorithm perpetually moves forwards through the sequence of collections, while ensuring that we always generate correctly according to the collection $\mathcal{C}_{j_t}$.

To see why $G$ non-uniformly generates with noise level $i$ for $\mathcal{C}$, let $K \in \mathcal{C}$ be an arbitrary target language. Since $\mathcal{C} = \bigcup_{j \in \mathbb{N}} \mathcal{C}_j$, there must exist a finite index $t_1$ such that $K \in \mathcal{C}_j$ for all $j \geq t_1$. Let $t^\star = \max(t_1, t^\star(\mathcal{C}_{t_1}))$. We claim that $G(x_0, \ldots, x_t) \in K \setminus S_t$ for all $t \geq t^\star$. First note that for all $t \geq t^\star$, we have $t^\star(\mathcal{C}_{t_1}) \leq t$ and $t_1 \leq t$. Thus, it must be that $j_t \geq t_1$, so $K \in \mathcal{C}_{j_t}$. Furthermore, since $t \geq t^\star(\mathcal{C}_{j_t})$, the output $G(x_0, \ldots, x_t) = G_{j_t}(x_0, \ldots, x_t)$ must be in $K \setminus S_t$, so $G$ must generate non-uniformly with noise level $i$ for $\mathcal{C}$. □

