# OpenReview forum: "Characterizing the Effect of Noise in Language Generation in the Limit"
_ICML.cc/2026/Conference — ICML 2026 regular_

### Official Review · Reviewer_Lqwa · 2026-03-05

**Soundness:** 4
**Presentation:** 4
**Significance:** 3
**Originality:** 3
**Overall Recommendation:** 5
**Confidence:** 1

**Summary:**

This paper presents a complete landscape of uniform and non-uniform noise-dependent generation. It establishes a surprising equivalence: in both settings, all finite noise levels ($i \ge 1$) possess the exact same generative capacity. This reveals a fundamental separation between noiseless generation and generation with even a single noisy string, highlighting the severe challenges noise introduces to language generation algorithms. Lastly, this work successfully delivers the first complete mathematical characterization for non-uniform noise-dependent generatability.

**Compliance With Llm Reviewing Policy:**

Affirmed.

**Final Justification:**

Thank the authors for their rebuttal. After considering the comments from other reviewers, I have decided to maintain my original score.

**Key Questions For Authors:**

See weaknesses.

**Limitations:**

yes

**Strengths And Weaknesses:**

Strengths:

The equivalence between different finite noise levels revealed in this paper is impressive, but I have no background in pure learning theory. **So the AC/other reviewers should ignore this review when making the final decision.**

Weaknesses:

It appears to rely on a strong assumption that the noise is strictly bounded by an absolute finite constant $i$.

As stated in the Introduction, *“Understanding the powers and limitations of Large Language Models (LLMs) is a fundamental problem in machine learning.”* In practice, however, LLMs often encounter more realistic and prevalent forms of noise, such as **proportional noise**. Examples include percentage-based label noise commonly observed in real training datasets, or persistent distribution shifts where the number of corrupted samples up to time $t$ may grow proportionally (e.g., at most $\epsilon \cdot t$).
Under such settings, it is unclear whether **Theorem 2.14** would still hold. Adding a section discussing these potential limitations and their relation to realistic noise patterns encountered by LLMs would help readers better understand the scope and applicability of the proposed theory.

---

> ### Author Rebuttal · Authors · 2026-03-31
>
> Thank you for the positive comments! The setting of proportional noise is very interesting and has been studied recently by Mehrotra et. al (2025). They define the density of noise to be the fraction of contaminated samples in the limit. They show that with $o(1)$ noise, i.e., an infinite but measure-zero fraction of samples are corrupted, all countable collections are generatable in the limit. For any constant $c > 0$, they give a complete characterization of collections that are generatable in the limit with noise density $c$, and show that for any $c > 0$, there is a finite collection that is not generatable.
>
> The definitions of uniform and non-uniform noise-dependent generatability considered in our paper allow the timestep $t^\star$ to depend on the specific (finite) noise level chosen by the adversary. It seems challenging to adapt this definition to the setting with $o(1)$ noise density, because it is unclear how we should allow the timestep to depend on the specific noise level. However, it is an interesting open question to explore which (uncountable) collections can be generated in the limit with $o(1)$ noise. We will include a discussion of this model in the camera-ready version.

---

> > ### Author Rebuttal · Reviewer_Lqwa · 2026-04-01
> >
> > I do not have other concerns

---

### Official Review · Reviewer_m9JB · 2026-03-11

**Soundness:** 3
**Presentation:** 3
**Significance:** 3
**Originality:** 3
**Overall Recommendation:** 4
**Confidence:** 2

**Summary:**

This paper improves the existing theory and understanding of language generation in the limit. In the scenario where an adversary inserts noisy strings, the paper shows that generation with a single noisy string is equivalent to generation with any number of noisy strings, and that this differs from the noiseless setup. This provides solutions to open questions posed by prior work.

**Compliance With Llm Reviewing Policy:**

Affirmed.

**Final Justification:**

The authors addressed my concerns.

**Key Questions For Authors:**

N/A

**Limitations:**

Yes

**Strengths And Weaknesses:**

## Strength:

1. The results are solid and advance our understanding in this direction.

2. The paper is well written.

## Weakness:

1. Citation formats are not used consistently or correctly between \citep and \citet. "Bai et al." and "Bai, Panigrahi, and Zhang (2026)" are referring to the same paper.

----

The reviewer has checked all theoretical results in the main paper and skimmed the appendix. However, the reviewer is not so familiar with the theory in this line of work and its related works, and therefore is limited in fully assessing the novelty relative to the broader body of prior results.

---

> ### Author Rebuttal · Authors · 2026-03-31
>
> Thank you for the positive comments! We will make sure that all of the citations are in a consistent format (using \cite) for the camera-ready submission.

---

> > ### Author Rebuttal · Reviewer_m9JB · 2026-04-01
> >
> > I do not have other concerns

---

### Official Review · Reviewer_BFZM · 2026-03-11

**Soundness:** 4
**Presentation:** 2
**Significance:** 3
**Originality:** 3
**Overall Recommendation:** 5
**Confidence:** 4

**Summary:**

This paper uses the “Language Generation in the Limit” framework to study the impact of noisy data on the classes of languages that an algorithm could learn to generate with a finite amount of samples. The language generation in the limit setting assumes a class of languages $\mathcal{C}$, and, for an arbitrary language $K \in \mathcal{C}$ in this class, assumes an algorithm $G$ which is iteratively given samples from that language $x_t \in K$ and subsequently outputs potential samples $z_t$. The observed samples $x_t$ are said to form an **enumeration** of the language $\mathbf{x} = x_1, x_2, ..., x_t, ...$. The framework then analyses whether there exists such an algorithm $G$ which can generate only samples in the language $z_t \in K$ after some finite amount of time $t^\star$. In particular, it further defines a class of languages $\mathcal{C}$ as **uniformly generatable** if there exists an algorithm $G$ and a finite constant number of samples $t^\star$, such that, for any language $L$ and enumeration $x$, all samples $z_t$ generated by the algorithm are a part of the language $L$ for $t>t^\star$. A similar definition exists for **non-uniformly generatable**, but the time $t^\star$ can depend on the language $L$. The paper’s main contributions are to show:
* That the classes of languages that are *uniformly generatable* given one noisy sample in enumeration $\mathbf{x}$, is the same as for any finite (but constant) number of noisy samples.
* That the classes of languages that are *non-uniformly generatable* given one noisy sample in enumeration $\mathbf{x}$, is the same as for any finite (but constant) number of noisy samples.
* That there  are classes of languages that are *uniformly generatable* given no noisy samples, but which are not uniformly generatable given 1 noisy sample.
* That there  are classes of languages that are *non-uniformly generatable* given no noisy samples, but which are not non-uniformly generatable given 1 noisy sample.

**Compliance With Llm Reviewing Policy:**

Affirmed.

**Final Justification:**

The authors addressed my main concerns. As I had already given this paper a relatively high score, I’m keeping it as is.

**Key Questions For Authors:**

These are not exactly questions, but actually comments or smaller pieces of feedback.

> Title: Quantifying the Effect of Noise in Language Generation

The titles in OpenReview (Quantifying the Effect of Noise in Language Generation) and in the pdf (Quantifying Noise in Language Generation) are slightly different, and I believe neither is ideal. No noise is actually quantified, so the title in the pdf, in my opinion, does not represent the paper’s contributions. The title in OpenReview is better, but I think without stating “in the Limit”, the title is still misleading, as it suggests the paper will analyse practical language generation scenarios, and not this “in the limit” setting. Personally, I’d change the title to something like “Characterising the Effect of Noise in Language Generation in the Limit” (with Characterising instead of Quantifying, and adding “in the Limit” to the title).

> Line 188, right column: The above equivalence is perhaps surprising, since it contrasts with the strict hierarchy for noisy generation in the limit shown by (Bai et al., 2026)—for each noise level i ≥0 there exists a collection which can be noisily generated in the limit with noise level i, but not noise level i + 1.

There are quite a few technical terms and subtle distinctions about it, which might be hard for a reader to grasp at first read and could thus be made explicit in the text. For instance, if I understand correctly, the mentioned results by Bai et al. (2026) are about a general “generation in the limit” property of language classes, while you analyse uniform and non-uniform generation in the limit. At first read, however, this paragraph “tripped” me, and I had to spend some time on it to understand the distinction between this prior work and yours.

**Limitations:**

I believe the authors could expand the discussion of how their results relate to practical results involving language generation using "real" language models.

**Strengths And Weaknesses:**

This paper analyses an interesting question: the effect of noisy observations on whether a language can be generated in the limit. The paper puts forward 4 new theoretical results regarding this formal learnability setting. However, while I don’t think this is necessarily critical, the paper’s results are purely theoretical, and the paper does not attempt to motivate the framework or its results with respect to practical language generation approaches based on state-of-the-art language models.

**Soundness.** The paper is, to the best of my knowledge, sound. I tried to check all the proofs, and they seemed correct. The literature review also seemed complete (although I am not a specialist in the "language generation in the limit" literature, so there could be important related work which I am not aware of).

**Presentation.**
The paper is well written, although the proofs could use a bit more hand-holding; in particular, there is a lot of importance in minor notational distinctions (e.g., subscripts indicating the noise level) which are hard to follow and keep track of.

**Significance.**
The paper tackles an interesting problem and puts forward new theoretical results regarding it. The paper's significance could be strengthened by connecting it to more practical settings, but I think the theoretical results are still valuable in themselves.

**Originality.**
The paper is original, to the best of my knowledge, and provides new interesting results.

---

> ### Author Rebuttal · Authors · 2026-03-31
>
> Thank you for the detailed feedback and positive comments! Regarding the presentation, we will include in the camera-ready version a description of the intuitive ideas behind the main proofs (especially Lemma 3.2 and Theorem 4.7).
>
> Your understanding is correct that Bai et al. (2026) study noisy generation in the limit, while we study noisy uniform and non-uniform generation. We will make this distinction more explicit and more clearly emphasize the contrast with prior work in the final submission.
>
> Thank you for the suggestions on the title. We agree that “Characterizing” is a more representative description than “Quantifying” and will change that in the camera-ready submission.

---

> > ### Author Rebuttal · Reviewer_BFZM · 2026-04-03
> >
> > I thank the authors for their response. Reading the title, I still believe that adding “in the limit” to the title is important:
> >
> > > I think without stating “in the Limit”, the title is still misleading, as it suggests the paper will analyse practical language generation scenarios, and not this “in the limit” setting. Personally, I’d change the title to something like “Characterising the Effect of Noise in Language Generation in the Limit” (with Characterising instead of Quantifying, and adding “in the Limit” to the title).
> >
> > At the moment, the title makes the paper contributions sound more general than they are. The paper is about the “in the limit” framework; while this is not an issue itself, the paper’s title should position it like that.

---

> > > ### Author Response · Authors · 2026-04-04
> > >
> > > Thank you for the response. We agree that “Characterizing the Effect of Noise in Language Generation in the Limit” better represents the paper and will make the change in the camera-ready version.

---

### Decision · Program_Chairs · 2026-04-30

**Decision:**

Accept (regular)

**Comment:**

This paper presents a complete landscape of uniform and non-uniform noise-dependent generation. It establishes a surprising equivalence: in both settings, all finite noise levels possess the exact same generative capacity. This reveals a fundamental separation between noiseless generation and generation with even a single noisy string, highlighting the severe challenges noise introduces to language generation algorithms. Lastly, this work successfully delivers the first complete mathematical characterization for non-uniform noise-dependent generatability.